# OpenReview forum: "Stance Elicitation as a Black-Box Framework for Auditing LLM Alignment"
_ICLR.cc/2026/Conference — Submitted to ICLR 2026_

### Official Review · Reviewer_StgR · 2025-10-29

**Soundness:** 2
**Presentation:** 2
**Contribution:** 2
**Rating:** 2
**Confidence:** 3

**Summary:**

This paper proposes a black-box framework called stance elicitation to audit how large language models (LLMs) are politically and morally aligned. Instead of looking inside the model, the method asks structured questions like “Would entity X agree with policy Y?” to map each model’s semantic space into measurable stances. The authors test several major LLMs across political and moral statements. The study also identifies systematic biases and gaps between stated and revealed alignment, plus demographic stereotyping.

**Strengths:**

1.The framework is simple, intuitive and works as a black-box audit.

2. It’s transparent and well-validated across multiple models.

3. The work bridges AI auditing and social science in a fresh, useful way.

**Weaknesses:**

1. The method mostly observes bias but doesn’t really explain or fix it.

2. There’s still a chance that the prompts themselves introduce bias — the paper doesn’t fully control for that.

3. The evaluation focuses on politics in Western contexts, which might limit generalization.

4. Some visual and statistical analyses (e.g., PCA plots) feel more descriptive than rigorous.

**Questions:**

Please see the weaknesses section — the paper’s presentation is difficult to follow and could benefit from further refinement, especially in improving clarity and figure readability.

---

> ### Author Response · Authors · 2025-11-12
>
> > The method mostly observes bias but doesn’t really explain or fix it.
>
> The position of the paper is that you cannot "fix" bias; any LLM necessarily ends up somewhere in the semantic space of political and philosophical positions so it is necessarily "biased". E.g. a position halfway between Democrats and Republicans in the US would simply occupy roughly the same spot as the German CDU.
>
> The paper rather identifies a number of other problems related to bias, namely: (1) frequent disagreement between stated and revealed positions, (2) strong disagreements between legal positions and LLM alignment (Figure 6b), and (3) implausible attributions of political positions to different demographics (Figure 6a), implying both racial/ethnic bias and incorrect statistics.
>
> > There’s still a chance that the prompts themselves introduce bias — the paper doesn’t fully control for that.
>
> The innovation of the paper is that it does not matter whether the prompts are "biased" since, unlike previous approaches, we do not score based on the responses directly. Rather, we use the totality of responses to a set of questions to derive a semantic space. Furthermore, the semantic spaces and maps are largely independent of the set of questions we choose; that is, even purely moral dilemmas still give us a similar political and philosophical spaces as policy-related questions.
>
> > The evaluation focuses on politics in Western contexts, which might limit generalization.
>
> The method makes no specific assumptions about Western contexts; it is directly applicable to other languages and cultures. For example, you could take a purely Chinese-language LLM, generate a policy set, and use Chinese political entities, figures, and philosophies as anchors. You could then even map Western parties into that Chinese space.
>
> > Some visual and statistical analyses (e.g., PCA plots) feel more descriptive than rigorous.
>
> The paper should be read as a paper about a novel, mathematically grounded visualization technique. I'm not sure in what way that is not "rigorous".
>
> TODO: Maybe make this clearer to reader.

---

### Official Review · Reviewer_Kcmj · 2025-10-29

**Soundness:** 2
**Presentation:** 2
**Contribution:** 3
**Rating:** 4
**Confidence:** 3

**Summary:**

The paper proposes a method based on stance elicitation for characterizing the political biases present in Large Language Models (LLMs). Specifically, the paper develops a series of prompts fundamentally consisting of an entity descriptor, a policy statement, and a yes/no question about the latter two components. The paper then uses the yes/no outputs from LLMs in order to characterize their political biases. It further compares the output biases of Pew Research Center’s political typology and RILE scores as a form of docking for validity. The paper also makes some observations about how some entities, like generic person descriptors, do not lead to reliable political biases in the outputs.

**Strengths:**

The paper addresses a significant problem and has some potentially significant results. First, characterizing the political bias present in LLMs, especially in a black box manner, is a societally important problem, as LLMs are increasingly being used to analyze political content and even simulate political opinions. Furthermore, results like the instability of alignment of personalities with political beliefs are an interesting result from a bias analysis standpoint and may explain some of the results seen in other LLM political bias papers (as pointed out in the paper).

**Weaknesses:**

There are some weaknesses in the grounding, novelty, and clarity of the paper. For the grounding of the paper, I think there are weaknesses both methodologically and empirically. For methodology its not clear how much the methodology of prompting the LLM to get numerical responses would lead to different results if perturbations were applied. For example its seem likely that perturbing the verbiage of elements like the entity and policies, even if the verbiage is roughly semantically equivalent but maybe with a different sentiment polarity, in the prompts could alter the responses. In Ng et al. “Examining the Influence of Political Bias on Large Language Model Performance in Stance Classification,” the authors found that differences in the datasets lead to downstream task performance differences. Additionally, how would results change if LLM were given a more nuanced way to evaluate positions of entities on policies, like a Likert scale versus a binary yes or no. Finally, the use of a tensor implies that each of the axes are all of the same thing along those axes. It's not clear from a qualitative perspective that policy issues are equivalent to something like moral principles; I would agree that they are related, but not that they are equivalent. Thus, it's not clear that a tensor is the right representation of the outputs of the models, given the entity and policy tests. From the empirical perspective, I think there are some issues with the completeness of the policy probes and presentation of some of the political viewpoints/parties. For the latter, in the U.S., the libertarian party is not a major party and does not represent a third pole of U.S. politics. In fact, most libertarians caucus and vote along Republican lines in major U.S. elections. For the former, while it's probably not possible to fully enumerate all policy positions to really find the contours of a party, there does not seem to be any grounding that the policy positions used in the prompting represent a reasonable basis to separate political parties.

For the novelty, it's not clear that the proposed method significantly improves upon the political surveying process done by Rozado in his works. The proposed method is, at its core, the same procedure Rozado uses (i.e., prompting LLMs on political questions and topics to evaluate their bias) but with a different structure around the prompts and a quantitative analysis of the outputs.

For clarity, I think the paper would really benefit from something like a flowchart to understand the full process of prompt generation $\rightarrow$ tensor $\rightarrow$ analysis of the tensor. I personally had a hard time tracing where the conclusions were coming from.

**Questions:**

1. What is the reason the PCA is used for nearly all projections except one (policies uses t-SNE) despite mentioning having used PCA, t-SNE, and UMAP?

2. How would results change if LLM were given a more nuanced way to evaluate positions of entities on policies? For example, what about using a Likert scale with degrees of agreement or disagreement?

---

> ### Author Response · Authors · 2025-11-12
>
> > "The proposed method is, at its core, the same procedure Rozado uses (i.e., prompting LLMs on political questions and topics to evaluate their bias) but with a different structure around the prompts and a quantitative analysis of the outputs."
>
> The proposed method is fundamentally and importantly different. Rozado uses a-priori political spaces with questions intended to map into that a-priori political space based on prior assumptions about the relationship between answers to those questions and the political space. Some of the spaces and questions Rozado uses (e.g., the Nolan test) have never been validated, and its dimensions and methodology are questionable.
>
> In contrast, our method does not assume any prior political space and makes no assumptions about the political implications of answers to any particular questions.  Instead, we construct a semantic space based on answers to questions, without any prior assumptions, axes, or human judgments. The political space and where entities map inside it are purely derived from the mutual relations of stance vectors.
>
> TODO: More explicitly contrast with Rozado in the introduction/conclusions to make this clearer.
>
> > "For example its seem likely that perturbing the verbiage of elements like the entity and policies, even if the verbiage is roughly semantically equivalent but maybe with a different sentiment polarity,"
>
> Unlike other methods, we do not score LLMs directly on their answers. Polarity and similar effects do not act as noise in our approach, but rather as useful and measurable signals of bias and differences between LLMs.
>
> For entities, the semantic shifts of seemingly semantically equivalent statements is something we actually want to measure to discover unexpected deviations and biases. So, for example, you can map a "legally compliant AI" relative to a "well-aligned AI" and observe that they are, in fact, different.
>
> TODO: Can we clarify this further in the paper?
>
> > "Additionally, how would results change if LLM were given a more nuanced way to evaluate positions of entities on policies, like a Likert scale versus a binary yes or no. "
>
> The LLMs are prompted and can respond with a floating point number between -1 and 1 (see A.3). In practice, responses tend to be strongly bimodal.
>
> > "Finally, the use of a tensor implies that each of the axes are all of the same thing along those axes. It's not clear from a qualitative perspective that policy issues are equivalent to something like moral principles;"
>
> The three axes of the stance tensor are Model x Entity x Policy, and each entry is just the LLM response in [-1,1]. There are separate stance tensors for the three different policy sets in the paper (principles policy set; legislation policy set; moral dilemma policy set); stance tensors for different policy sets are not mixed because they are indeed not comparable.
>
> > "For the former, while it's probably not possible to fully enumerate all policy positions to really find the contours of a party, there does not seem to be any grounding that the policy positions used in the prompting represent a reasonable basis to separate political parties."
>
> Unlike Rozado, we are not purposely designing questions to separate political parties. We are mapping semantic spaces and see where different entities fall. It is an empirical result of our work that parties are, in fact, (1) reliably and consistently separated by many different policy sets, and (2) reliably and consistently closely align with particular philosophical positions (Democrats-Rawls, Libertarians-Nozick, Republicans-Nozick/Burke).
>
> > "For the latter, in the U.S., the libertarian party is not a major party and does not represent a third pole of U.S. politics. In fact, most libertarians caucus and vote along Republican lines in major U.S. elections."
>
> Rozado presupposes a 2D political space including "Libertarian". In our work, the 2D political space is an _empirical_ result, with D/R/L spanning the 2D space, not just within the US but also internationally. And as Figure 3(a) shows, ideologically and semantically, Republicans and Libertarians do indeed map closely together. Unlike Rozado's a priori maps, the empirically derived semantic space is roughly triangular. And as Figure 6b shows, the Libertarian/Nozick pole is very important for alignment/legal mapping.
>
> TODO: Maybe comment on this as well.
>
> > "What is the reason the PCA is used for nearly all projections except one (policies uses t-SNE) despite mentioning having used PCA, t-SNE, and UMAP?"
>
> Three reasons: (1) PCA accounts for >80% of the variance in all cases so it is a good representation, (2) PCA preserves spatial relations and lets us meaningfully talk about "halfway between Nozick and Burke", and (3) PCA allows us to project onto entity-selected subspaces (US parties in German political space vs German parties in US political space).

---

### Official Review · Reviewer_y9FM · 2025-11-02

**Soundness:** 1
**Presentation:** 1
**Contribution:** 1
**Rating:** 0
**Confidence:** 4

**Summary:**

The paper proposes a “stance elicitation” method to audit LLM alignment. It builds a stance tensor from model responses to policy statements and uses PCA/t-SNE to project ideological positions. The authors claim it reveals consistent political and philosophical structures and highlights gaps between stated and revealed alignments. However, the setup and evidence are informal and mostly qualitative.

**Strengths:**

* The stance tensor idea is a clear, general way to visualize model positions.
* The generic vs. rule-based contrast is interesting for alignment diagnostics.
* Some visual patterns (party clusters, philosophy alignment) seem intuitive and interpretable.

**Weaknesses:**

* No robustness checks or variance across seeds/prompts; results may not be stable.
* Heavy reliance on visualizations without statistical testing or quantitative evaluation.
* Lacks comparison to recent political bias auditing baselines.
* Code and details are missing; the paper also doesn’t follow ICLR formatting.

**Questions:**

1. The paper claims to reveal political and philosophical structures “more faithfully than prior auditing approaches” (Sec. 1, Conclusion). Which prior methods are these, and why are there **no quantitative comparisons** to established frameworks such as Bang et al. (2024), Argyle et al. (2023), or Motoki et al. (2025)?
2. None of the figures include **error bars, variance, or statistical tests**. How stable are the findings across random seeds, temperatures, or prompt paraphrases?
3. Several evaluation items were reportedly **generated or validated by the same model families** later audited. How do the authors rule out circularity or data leakage?
4. The RILE correlation (r = 0.99) seems implausibly high without sample size or CIs. How many parties were included, and is this robust to translation and summarization?
5. Why does the submission **not use the official ICLR template**? Was this intentional?
6. Will a compliant version and code with fixed API details be released for reproducibility?

---

> ### Author Response · Authors · 2025-11-12
>
> > No robustness checks or variance across seeds/prompts; results may not be stable.
>
> As the paper states: "Unless otherwise stated, results are reported on the ‘‘political principles’’ set, but results are broadly consistent across all four sets. Statistical noise in model responses falls into two categories: sampling noise based on the size of the policy set, and stochastic responses from the model; both can be demonstrated to be smaller than the effects we are measuring, but we leave out such an analysis for brevity in this paper."  (Error bars due to sampling noise are about 5%, to give some idea of the order of magnitude.)
>
> In addition, we have done extensive testing with numerous different policy sets, prompts, and entity descriptions, all yielding similar results, but a presentation and evaluation of such results would go far beyond the scope of this paper. This paper simply introduces the method and shows that it yields useful semantic maps.
>
> > Heavy reliance on visualizations without statistical testing or quantitative evaluation.
>
> That's because the paper is a paper about visualizing semantic spaces, not about testing a hypothesis or optimizing a score.
>
> TODO: Maybe change the title to make this clearer; "A Stance Tensor Method for Visualizing the Semantic Spaces of LLMs"?
>
> > Lacks comparison to recent political bias auditing baselines.
>
> The paper is not concerned with deriving a particular political bias score; in fact, as the paper explains, we consider such 1D scoring to be not particularly useful. Instead, the paper is concerned with visualizing semantic spaces in order to understand LLM _alignment_ in more detail.  (The paper derives a left-right score and correlates it with the RILE score to show that the visualizations contain information that closely corresponds to expert judgment, but deriving left-right scores is not a primary objective of the paper.)
>
> The most important alignment-related results are the substantial deviations between stated/revealed legal constraints and stated/revealed alignment (Figure 6b), a potentially serious problem for LLMs. The divergent attributions of political positions to demographics (Figure 6a) are also a significant concern. Other results: the close clustering of "good", "expert", "aligned", and "human " (Figure 3b) with Rawls/D is interesting, as is the alignment between political parties and political philosophies (Figure 3a), previously largely unrecognized in this field. None of these have been previously recognized.
>
> Response to questions:
>
> 1. Bang et al. (2024), Argyle et al. (2023), or Motoki et al. (2025) measure different things and do not constitute baselines for this work; our framework measures something qualitatively different and has different objectives. For benchmarking, we view the expert-derived RILE scores as authoritative.
>
> TODO: Can we make this point more strongly?
>
> 2. It is unclear what "error bars" or "CI" the reviewer is asking for. There is some sampling error due to the choice of policy questions (<5%). But the purpose of the paper is to visualize the semantic spaces of different LLMs under different conditions. These maps are highly reproducible under different choices of questions, policy sets, and well-specified entities, but we cannot show all those results within the space constraints of this conference.
>
> TODO: Strengthen the disclaimer.
>
> 3. This is not a quiz where there are right or wrong answers, so there is nothing to "leak". LLMs were simply used to generate statements of moral dilemmas, philosophical principles, and politically important issues (examples in the appendix). Unlike other methods, we do not score based on whether LLMs agree or disagree with a policy, we score on how LLMs respond relative to each other, avoiding the problems with polarity or bias previous methods had. Many of the "policy" statements are not even related to political questions.
>
> TODO: Maybe try to clarify this even more in the paper.
>
> 4. The information for how the RILE score was computed is given in the paper: this is the result of a 1D projection of 100 response vectors from four language models (an error bar on this would be about 2.5%, which is why we don't show it in any of the plots). Yes, the correlation is amazingly high, in particular given that the way we derive this score is completely different from the way human experts compute this score. We did not tune or manipulate this, this is simply the result we got (and we would have preferred a slightly lower correlation to avoid this discussion). Again, the left-right score is not an objective of our paper in the first place, it is simply evidence that the semantic spaces we derive closely correspond to human intuition and ratings.
>
> 5. We are not aware of any deviations from the ICLR template; the paper uses the LaTeX template and format checkers didn't flag any errors. Please indicate any deviations you see.
>
> 6. As stated in the paper, the code will be released in open source form.

---

### Author Response · Authors · 2025-11-21
**significant revision clarifying the contributions**

We want to thank the reviewers for their input and feedback. We have substantially revised the paper to clarify and address the issues raised by the reviewers.

Many of the issues raised were a result of the paper not being clear enough about its objectives, namely that of visualizing and understanding semantic spaces, rather than measuring alignment with pre-defined metrics. It is an empirical result of the paper that some of the semantic spaces are well-aligned with human ground truth.

We still have not incorporated error ellipses into the plots because they would only represent sampling error and are of the order of a few percent. We could regenerate the figures with error ellipses, but that is going to take a little longer.

(We understand that the revised version is longer and will need to be shortened if accepted.)

---

### Meta-Review · Area_Chair_7Kv5 · 2026-01-07

**Summary:**

This paper introduces a stance-tensor method to visualize the semantic spaces of large language models, aiming to analyze their responses to structured entity-policy queries, reveal intrinsic political and moral structures, and quantify semantic inconsistencies for auditing and alignment purposes.

The paper received one strong reject, one borderline reject, and one reject. In their reviews, the reviewers identify several major issues that undermine the significance of the work. In particular, reviewers question whether the observed structures reflect genuine model behavior, as sensitivity to prompt wording, sentiment, and decoding settings is not evaluated (y9FM, Kcmj, StgR). There are also conceptual and empirical concerns regarding the validity of the proposed representations and political probes, including incomplete grounding of policy and party choices and a narrow focus on Western political contexts (Kcmj, StgR). Finally, claims of novelty and improved faithfulness over prior political bias auditing work are not substantiated by quantitative comparisons to established baselines (y9FM, Kcmj).

**Reviewer Concerns:**

The authors provide some intuitive clarifications in the rebuttal. However, in the AC's view, without providing sufficient quantitative evidence, these clarifications are insufficient to resolve the core concerns raised by the reviewers.

**Reviewer Scores:**

The AC believes that the reviewers would likely maintain their original scores, and that the paper remains well below the acceptance threshold.

---

### Decision · Program_Chairs · 2026-01-26

Reject